# Progress in Investigating the Impact of Obesity on Male Reproductive Function

**DOI:** 10.3390/biomedicines13092054

**Published:** 2025-08-23

**Authors:** Yafei Kang, Peiling Li, Suying Yuan, Sen Fu, Xue Zhang, Jiaxing Zhang, Chenle Dong, Renhui Xiong, Hu Zhao, Donghui Huang

**Affiliations:** 1Institute of Reproductive Health, Tongji Medical College, Huazhong University of Science and Technology, Wuhan 430030, China; d202381936@hust.edu.cn (Y.K.); m202476185@hust.edu.cn (S.Y.); 19115123432@163.com (S.F.); m202376031@hust.edu.cn (X.Z.); u202210285@hust.edu.cn (J.Z.); dongchenle@163.com (C.D.); u202210281@hust.edu.cn (R.X.); 2Tongji Medical College, Huazhong University of Science and Technology, Wuhan 430030, China; peilingli0428@163.com; 3Department of Human Anatomy, Tongji Medical College, Huazhong University of Science and Technology, Wuhan 430030, China; 4National Demonstration Center for Experimental Basic Medical Education, Huazhong University of Science and Technology, Wuhan 430030, China

**Keywords:** obesity, white adipose tissue, brown adipose tissue, sperm quality, epigenetics, male infertility

## Abstract

Obesity represents a significant global public health challenge, which not only elevates the risk of mortality but also increases the likelihood of chronic diseases. The ongoing obesity epidemic has led to a growing recognition of the detrimental effects of excessive adipose tissue accumulation on male reproductive health. Substantial evidence indicates that obesity adversely affects sperm quality, thereby impairing male fertility. Specifically, obesity is associated with compromised spermatogenesis, erectile dysfunction, and detrimental effects on offspring fertility parameters. These effects are mediated through various mechanisms, including alterations in the hypothalamic–pituitary–gonadal axis, inflammation within the reproductive system, localized caloric excess in reproductive tissues, epigenetic modifications, disruptions in gut microbiota, and heightened oxidative stress levels. While the molecular alterations associated with obesity have been extensively documented, the precise mechanisms by which obesity influences male reproductive function remain inadequately understood. This article aimed to review the classification and distribution of adipose tissue in obesity, the impact of obesity on male fertility, and the potential mechanisms through which obesity affects male reproductive health, thereby offering insights into the prevention and treatment of obesity-related male fertility issues.

## 1. Introduction

Male infertility is defined by the World Health Organization (WHO) as the inability of a male to impregnate a fertile female after at least one year of regular, unprotected sexual intercourse. Males are solely responsible for approximately 20% of infertility cases and contribute to an additional 30% to 40% of cases overall [1]. Numerous causes and risk factors underlie the rising incidence of male infertility, which can be categorized as congenital, acquired, or idiopathic. Approximately 30–50% of male infertility cases are idiopathic, with no identifiable cause or associated female infertility [2]. Environmental and occupational exposure to toxic chemicals [3], as well as various lifestyle factors, including smoking [4,5], alcohol consumption [6], obesity [7], and psychological stress [8,9] have all been identified as potential risk factors for male infertility.

Obesity is a complex disorder arising from the interaction of genetic, environmental, and psychosocial factors and is defined as a syndrome characterized by excessive accumulation of visceral fat [10]. Over recent decades, obesity has become a global epidemic. An analysis of data from 67.8 million individuals across 195 countries between 1980 and 2015 demonstrated that the prevalence of overweight among children and obesity among adults has doubled in more than 70 countries, with a continuing upward trend observed in most others [11]. Globally, approximately 1.9 billion adults (≥18 years) are classified as overweight, representing 39% of the adult population, including over 650 million individuals who meet the criteria for obesity (13%) [12]. Projections suggest that by 2025, obesity prevalence may increase to 18% among males and exceed 21% among females, while severe obesity could affect more than 6% of men and 9% of women worldwide [13].

Obesity represents a complex disorder with multiple contributing factors, encompassing nutritional patterns, sleep deprivation, exercise frequency, pharmacological influences, hereditary factors, and familial tendencies. It significantly elevates the risk of cardiovascular diseases and is strongly associated with numerous comorbid conditions, such as type 2 diabetes mellitus and several forms of cancer [14]. Moreover, obesity, recognized as a chronic health condition, is linked to a spectrum of complications encompassing mental health disorders, cardiovascular disease, metabolic syndrome, cancer, and increased mortality rates [15,16,17]. Recent literature has increasingly focused on the detrimental effects of obesity on male reproductive function. Empirical evidence indicates that elevated body mass index (BMI) is correlated with impairments in sperm quality, viability, and hormonal profiles, which may contribute to male infertility [18]. The mechanisms by which obesity adversely affects semen quality and male fertility include detrimental changes in semen parameters [19,20], endocrine disruptions [21,22], reproductive system inflammation, and heightened oxidative stress [23,24,25]. Furthermore, epigenetic studies suggest that the metabolic consequences of male obesity may induce epigenetic modifications during embryogenesis, potentially impacting the somatic health of offspring through alterations in epigenetic markers within their somatic tissues [26]. Accordingly, this review systematically explores the classification of adipose tissue, its anatomical distribution, and the pathophysiological effects of obesity on male reproductive function.

## 2. Classification and Distribution of Adipose Tissue

Adipose tissue is present in two functionally and structurally distinct forms: white adipose tissue (WAT) and brown adipose tissue (BAT) [13]. WAT, the primary fat depot in humans, is composed of adipocytes characterized by unilocular lipid droplets, relatively sparse mitochondria, and a substantial capacity for energy storage [27]. Anatomically, WAT is divided into two main depots: subcutaneous fat, located beneath the skin in regions, such as the thighs, buttocks, lower abdomen, and pubic area, and visceral fat, which surrounds internal organs [28]. The principal metabolic function of WAT is the storage of triglycerides to maintain energy homeostasis, serving as a fuel reserve during fasting or periods of increased energy demand [13]. Beyond its role as the main fat storage depot, WAT also functions as the largest endocrine organ, capable of secreting various adipokines and cytokines [29]. Adipokines are considered a critical link between obesity and infertility; for example, lipocalin, an adipokine, has been shown to positively influence sperm parameters, whereas other adipokines, including leptin, resistin, and chemotactic factors, may negatively impact spermatogenesis [30].

BAT predominantly localizes to specific anatomical regions, including the cervical, perirenal, suprarenal, and cardiothoracic areas, particularly surrounding the aorta and mediastinal structures [31]. Histologically, brown adipocytes are characterized by small, multilocular lipid droplets, in contrast to the unilocular lipid droplets characteristic of white adipose tissue. Additionally, BAT contains more mitochondria than WAT, which are characterized by dense, brownish cristae due to the presence of ferric heme-containing cofactors in cytochrome oxidase. This structural distinction suggests an enhanced capacity for oxidative metabolism [32]. Table 1 illustrates the distinct differences between WAT and BAT. The primary function of BAT is to generate heat through non-shivering thermogenesis, thereby protecting against cold-induced hypothermia without eliciting shivering responses. This thermogenic function is mediated by mitochondrial uncoupling protein 1 (UCP1) in BAT, which uncouples oxidative phosphorylation in the mitochondrial inner membrane, leading to heat production and increased energy expenditure [33]. Furthermore, research indicates that estrogen can modulate BAT activity by upregulating thermogenesis, and increased BAT activity is associated with elevated body temperature, which inversely correlates with body weight [34].

Beyond the classical classifications of white and brown adipose tissues, a distinct third adipocyte population, termed beige adipose tissue, has been characterized. These adipocytes are predominantly situated within subcutaneous fat depots, notably in the paravertebral and supraclavicular regions. Although histologically indistinguishable from brown adipocytes, beige adipocytes exhibit physiological properties similar to white adipocytes under basal conditions, demonstrating limited thermogenic activity [38]. Importantly, exposure to cold environments and physical exercise can trigger the transdifferentiation of white adipocytes into beige adipocytes through a process referred to as “browning,” during which the cells acquire morphological and metabolic features characteristic of brown fat [39]. Upon exposure to reduced ambient temperatures, beige adipocytes can promptly and dynamically activate thermogenic pathways via adaptive responses, thereby enhancing energy expenditure. Notably, once the browning stimulus is removed, beige adipocytes swiftly revert to their white adipose state [40]. Consequently, beige adipocytes represent a promising therapeutic target for obesity by promoting increased energy dissipation. The anatomical distribution of adipose tissue within the human body is illustrated in Figure 1.

The testes constitute vital components of the male reproductive system, primarily responsible for the generation of spermatozoa and the production of the sex hormone testosterone [41]. Following their release from the testes, spermatozoa exhibit limited motility and lack the ability to recognize or fertilize oocytes. The epididymis functions as a dynamic organ that facilitates the maturation of sperm, a process modulated by androgens. Both spermatogenesis and sperm maturation are highly temperature-sensitive, necessitating an environment slightly below the core body temperature [42]. Consequently, the distribution and quantity of adipose tissue within the male reproductive system may significantly influence the functional capacity of male reproductive organs.

Extensive research has established that sexually dimorphic patterns of adipose tissue distribution in mammalian species, including humans, are closely associated with distinct metabolic risk profiles [43]. Specifically, comparative analyses reveal that females predominantly accumulate subcutaneous fat in the gluteofemoral region, whereas males tend to deposit visceral adipose tissue within the abdominal cavity [44]. Similar to humans, adipose tissue in rodents functions as a multifunctional organ comprising multiple fat depots. In rodents, visceral fat pads surrounding the gonads are designated as epididymal fat in males and periovarian fat in females [45]. The epididymal adipose tissue (EAT) represents a specialized abdominal fat compartment characterized by unique anatomical and metabolic properties. This bilaterally symmetrical fat depot is situated caudal to the respective testis and epididymis, extending rostrally toward the diaphragm. Anatomically, it is associated with the spermatic cord vessels, which it accompanies medially into the retroperitoneal space. Based on its spatial relationship to the spermatic vessels, epididymal fat can be subdivided into three distinct zones: (i) the medial zone, which surrounds the spermatic artery and its main branches; (ii) the caudal zone, attached to the testis and epididymis; and (iii) the rostral zone, representing the proximal, flaccid extremity of the medial zone [46]. Chu et al. demonstrated that surgical excision of epididymal white adipose tissue (eWAT) in rodent models led to significant reductions in both local epididymal fat mass and total white adipose tissue [47]. Furthermore, this excision was associated with a decline in spermatogenesis, indicating that the presence of growth and/or trophic factors essential for normal spermatogenesis in eWAT is closely linked to male fertility; however, this relationship has yet to be thoroughly investigated.

## 3. The Impact of Obesity on Male Fertility

### 3.1. The Impact of Obesity on Spermatogenesis and Sperm Quality

Obesity has been associated with impaired sperm quality and diminished male fertility [48]. A meta-analysis involving large samples indicated that individuals with a BMI classified as obese or overweight exhibited significantly reduced semen quality [49]. Furthermore, an investigation into the factors influencing the DNA fragmentation index (DFI) of spermatozoa in a cohort of 1010 men affected by infertility in China revealed a positive correlation between DFI and the levels of triglycerides and total cholesterol in seminal plasma [50]. In studies involving HFD (high fat diet)-induced obese mice, a marked decrease in sperm viability was observed in the caudal region of the epididymis, with the percentage of immobile spermatozoa reaching as high as 60% [51]. Additionally, research by Jing et al. demonstrated that HFD exposure disrupted mitochondrial function in sperm and increased oxidative stress levels, which subsequently led to a reduction in spermatozoa viability [52]. Oliveira et al. proposed that the accumulation of adipose tissue in the suprapubic and scrotal regions resulted in elevated testicular temperatures, thereby contributing to the impairment of spermatogenesis [53]. Moreover, Komninos et al. reported that the testes of HFD-induced obese knock-out Leiden mice exhibited abnormal organization of seminiferous tubules at critical stages (VII and VIII) and alterations in the spermatogenic cycle [54]. Semen samples collected from overweight and obese men exhibited a markedly decreased sperm concentration, a reduced fraction of spermatozoa displaying progressive motility, and a lower percentage of spermatozoa with normal morphological characteristics relative to samples obtained from lean men [55].

### 3.2. The Impact of Obesity on Erectile Dysfunction

Erectile dysfunction (ED), that is, the consistent failure to achieve/maintain erection sufficient for intercourse, demonstrates strong epidemiological links with obesity. A clinical study conducted by Aleksandra et al. demonstrated significant correlations between the severity of ED, parameters of body fat accumulation, and gonadal hormone profiles in elderly male subjects aged 60 years and above [56]. Their findings suggest that excessive visceral fat accumulation contributes to the pathogenesis of ED. Additionally, two cross-sectional analyses utilizing data from the National Health and Nutrition Examination Survey (NHANES) between 2001 and 2004 identified the visceral obesity index as an independent risk factor for ED. Elevated visceral fat metabolism scores were also found to have a nonlinear association with increased ED prevalence [57,58]. Emerging evidence positions visceral adiposity as a critical predictor of ED, with clinical investigations revealing strong correlations between metabolic parameters and ED risk [59]. The pathophysiology of ED extends beyond aging to include obesity-related metabolic dysfunction, as supported by both epidemiological data in humans and experimental models involving HFD-induced obesity in rodents [60]. A comprehensive clinical study by Kaya et al. reported that 79% of ED patients were overweight or obese (BMI ≥ 25 kg/m^2^), with obese individuals (BMI > 30 kg/m^2^) exhibiting a 30% higher risk of sexual dysfunction compared to normal-weight controls [61]. These collective findings underscore the critical role of metabolic health in male sexual function, with visceral adiposity serving as both a biomarker and potential therapeutic target [62]. Therapeutic interventions, including structured physical activity and controlled dietary regimens, have demonstrated efficacy in ameliorating ED by addressing underlying risk factors and directly improving clinical outcomes [63]. Furthermore, surgical weight reduction procedures have been shown to significantly restore erectile function in affected individuals [64].

### 3.3. The Impact of Male Obesity on Offspring

Excessive consumption of an HFD by parents before, during, and after lactation may adversely affect the reproductive health of male offspring [65]. Substantial evidence indicates that diet-induced paternal obesity is significantly associated with impaired embryonic development. Moreover, paternal obesity has the potential to negatively influence embryos at various developmental stages, leading to related complications. Studies using animal models have demonstrated that paternal obesity exerts considerable detrimental effects on embryos during critical early developmental phases, including delayed oocyte fertilization, disruptions in cell cycle progression during the second and third cleavage stages, and reduced blastocyst growth. These factors may contribute to delayed zygote development, decreased placental size, and smaller offspring [66]. Bernhardt et al. reported that paternal obesity is linked to the upregulation of *Gata6* and *Samd4b* gene expression in embryos, suggesting that paternal obesity may induce alterations in male germ cells that correspond with subsequent changes in gene expression in preimplantation embryos [67]. Emerging evidence also indicates that paternal obesity is associated with morphological abnormalities in spermatozoa within seminiferous tubules, along with reduced sperm quality parameters in F1 offspring [68].

## 4. Mechanisms Through Which Obesity Impacts Male Fertility

The underlying mechanisms are primarily linked to dysfunction of the hypothalamic–pituitary–gonadal axis, inflammation within the reproductive system, increased localized heat in the reproductive organs, epigenetic modifications, disturbances in gut microbiota, and heightened levels of oxidative stress (Figure 2).

### 4.1. Endocrine Disorder

Current epidemiological data indicate that approximately 45% of adult males meet the criteria for moderate-to-severe obesity, a population characterized by a high prevalence of endocrine dysfunction, including abnormalities in sex hormone levels [69]. Of particular clinical importance, obesity-related secondary hypogonadism constitutes one of the most common and impactful endocrine disorders within this patient group [70].

The proper functioning of the hypothalamic–pituitary–gonadal (HPG) axis is critically regulated by the rhythmic secretion of gonadotropin-releasing hormone (GnRH). This neuroendocrine control system is highly sensitive to metabolic status and energy balance. When energy intake surpasses physiological requirements, this delicate equilibrium is disrupted, initiating a cascade of reproductive consequences, including hormonal imbalances, testicular oxidative damage, and impaired regulation of adipokines. These metabolic disturbances significantly affect gonadal physiology, secondary reproductive structures, and early embryonic development [71,72]. Expansion of adipose tissue promotes upregulation of adipocyte aromatase, thereby enhancing the peripheral conversion of androgens to estrogens, particularly the conversion of testosterone to 17β-estradiol. This estrogenic microenvironment exerts negative feedback on the hypothalamic–pituitary–testicular (HPT) axis, suppressing GnRH pulsatility and consequently reducing the secretion of luteinizing hormone (LH) and follicle-stimulating hormone (FSH) secretion [73]. Furthermore, studies utilizing animal models have demonstrated that Kiss1 neurons, when expressing leptin receptors, may play a crucial role in conveying metabolic information to gonadotropin-releasing hormone (GnRH) through the release of kisspeptin. Consequently, these neurons serve as significant modulators of GnRH/LH/FSH release, and diminished leptin signaling is associated with a reduction in GnRH neuronal activity [74]. Therefore, in the context of obesity, a decrease in hypothalamic kisspeptin expression may inhibit the pulsatile release of hypothalamic GnRH, resulting in hypothalamic hypogonadism [75].

### 4.2. Inflammation of the Reproductive System

Obesity constitutes a state of chronic low-grade systemic inflammation mediated by various pro-inflammatory factors [76]. Experimental investigations have shown that consumption of HFD induces activation of the NLRP3 inflammasome and subsequent pyroptotic cell death within testicular tissue, thereby impairing spermatogenesis in obese animal models [77]. This inflammatory environment is sustained by elevated secretion of pro-inflammatory cytokines, establishing a vicious cycle of metabolic dysfunction and reproductive impairment [78]. Within the epididymal environment, this pro-inflammatory state disrupts the function of epididymal epithelial cells and enhances the infiltration of neutrophils and macrophages into the epididymal lumen. This infiltration results in the hyper-expression of cytokines and apoptosis of epithelial cells, which hinders spermatogonial maturation and fertilization [79]. Mu et al. demonstrated that reactive oxygen species (ROS) generation is a critical factor in NLRP3 activation, which facilitates the secretion of interleukin-1 beta (IL-1β) by supportive testicular cells [80]. This mechanism adversely affects testosterone synthesis and sperm function, activates nuclear factor kappa-light-chain-enhancer of activated B cells (NF-κB), and upregulates matrix metalloproteinase-8 (MMP-8) expression, resulting in occludin degradation and consequent disruption of the blood–testis barrier integrity and spermatogenesis. Schjenken et al. reported reduced concentrations of key immunoregulatory factors in the seminal vesicle fluid of males fed a high-fat diet, notably decreases in transforming growth factor-beta 1 (TGF-β1), interleukin-10 (IL-10), and tumor necrosis factor (TNF) [81]. The study also documented significant alterations in cellular populations, including a 27% reduction in CD4+ T cells, a 26% decrease in FOXP3+CD4+ regulatory T cells (Tregs), and a 19% decline in CTLA4+ Treg subsets. These results demonstrate that diet-induced obesity modifies the immunoregulatory profile of seminal plasma, reducing its ability to promote tolerance-inducing immune reactions in female reproductive tissues.

Chaudhuri et al. demonstrated that the primary pro-inflammatory cytokines secreted by adipose tissue, specifically interleukin-6 (IL-6) and tumor necrosis factor alpha (TNF-α), directly stimulate the hypothalamic–pituitary–adrenal (HPA) axis [82]. This stimulation leads to increased cortisol secretion alongside the suppression of thyrotropin and testosterone secretion, thereby promoting visceral fat accumulation and dysfunction of the hypothalamic–pituitary–gonadal (HPG) axis. Moreover, obesity induces systemic inflammation, including a localized inflammatory response within the male reproductive tract. Such inflammation disrupts critical regulatory processes involved in spermatogenesis and sperm maturation, ultimately resulting in reduced male fertility. Wang et al. reported that obesity is associated with deficiencies in steroidogenesis-related enzymes essential for testosterone synthesis, further exacerbating these effects [77]. Additionally, Lainez et al. suggested that inflammation-induced synaptic remodeling may contribute to hypothalamic damage, leading to decreased levels of gonadotropins, testosterone, and sperm counts, with these effects being more pronounced in males than in females [83].

### 4.3. Elevated Localized Heat in the Reproductive System

Spermatogenesis is widely recognized as a process highly sensitive to temperature variations within reproductive physiology, requiring an optimal thermal range of 32–35 °C for proper function in the human testis. Obesity has been associated with increased body temperature, particularly in the scrotal region [66]. Furthermore, extragonadal heat production has emerged as a significant concern in obese individuals, primarily due to increased scrotal adiposity, as well as augmented fat deposits in the suprapubic and thigh regions [84]. In obese conditions, the accumulation of scrotal fat correlates with elevated levels of ROS [85]. Garolla et al. reported significantly higher scrotal temperatures in both varicocele patients and obese males compared to healthy controls [86]. This disruption in thermal regulation adversely affects testicular physiology, thereby impairing spermatogenic efficiency and reproductive capacity. Notably, human spermatogenesis demonstrates remarkable temperature sensitivity–optimal sperm production requires a testicular environment 2–4 °C below core body temperature, with each 1 °C elevation above this threshold reducing spermatogenic output by approximately 14% [87]. Consequently, the rise in scrotal temperature associated with obesity results in an increase in testicular temperature, underscoring the importance of maintaining the testes within the optimal temperature range to preserve healthy sperm production and overall fertility.

### 4.4. Epigenetic Modification

During spermiogenesis, the terminal phase of sperm development, extensive post-translational modifications, including histone acetylation and ubiquitination, mediate the critical transition from nucleosomal to protamine-based chromatin packaging. These biochemical modifications induce chromatin decondensation, enabling the systematic displacement of histones by protamines—an essential transformation that permits extreme DNA compaction within the mature sperm head [88]. Conversely, environmentally induced epigenetic mechanisms have been implicated in the manifestation of an obese phenotype, which is linked to the dysregulation of adipokines and subsequent reproductive dysfunctions [89]. Fofana et al. demonstrated that histone acetylation and ubiquitination, as well as the levels of protamine 1 (prm1), were significantly lower in the sperm cells of obese mice compared to controls [90]. Furthermore, obesity in male mice was found to inhibit the replacement of histones with protamines, rendering sperm DNA more vulnerable to damage from external factors, thereby adversely affecting sperm quality.

Although the nutritional and health status of the mother during pregnancy and lactation exerts a more pronounced influence on offspring health compared to that of the father, the paternal contribution remains significant and should not be overlooked [91]. Epigenetic research indicates that the metabolic imprint associated with obesity in men at the time of conception may lead to epigenetic modifications that can be inherited across generations or directly to the offspring via sperm. This process alters epigenetic markers in the somatic cells of the offspring, potentially exerting a significant impact on their physical health [26]. Abnormal sperm parameters, resulting from altered testicular metabolic profiles due to early life exposure to an HFD, have been shown to persist for up to two generations in murine models, illustrating a phenomenon referred to as “metabolic genetic memory” of HFD exposure [92]. Such epigenetic modifications, commonly referred to as epigenetic marks, primarily involve DNA methylation, histone modifications, and small non-coding RNAs [93].

#### 4.4.1. Impact of Obesity on Sperm DNA Methylation

Recent research has revealed that obese males exhibit distinct patterns of sperm DNA hypomethylation, particularly at imprinted loci and repetitive genomic regions, which are associated with reduced reproductive outcomes [94]. This epigenetic dysregulation is hypothesized to contribute to the paternal metabolic programming of offspring phenotypes through multiple molecular mechanisms. Specifically, obesity-related alterations in sperm epigenetic markers—including DNA and histone modifications as well as non-coding RNAs—may disrupt paternal genomic imprinting and transcriptional regulation during critical stages of embryonic development [95].

Animal studies have demonstrated that diet-induced paternal obesity in F0 mice alters the microRNA composition of sperm and modifies the methylation patterns of germ cells. Notably, a reduction in overall germ cell methylation has been observed in the F2 generation offspring, indicating its potential as a biomarker for assessing offspring health and the heritable transmission of obesity and metabolic dysfunction across generations [96]. In human studies, Keyhan et al. detected 3264 BMI-associated CpG methylation sites in human sperm, with significant enrichment in transcriptional regulatory genes and pathways involved in oncogenesis, neural development, and the maintenance of stem cell pluripotency [97]. Donkin et al. performed an extensive epigenomic comparison between sperm samples from obese and lean individuals, identifying significant alterations in both small non-coding RNA expression profiles and DNA methylation landscapes [98]. Notably, their investigation revealed that bariatric surgery-induced weight loss in severely obese males triggered methylation changes in 1509 genes within just seven days post-operation, demonstrating the dynamic nature of epigenetic reprogramming during late spermatogenesis. Wu et al. reported consistent methylation signatures at the Igf2/H19 imprinting control region across both hepatic tissue and spermatozoa in offspring of obese fathers, providing direct evidence for transgenerational epigenetic inheritance through the paternal germline [99].

#### 4.4.2. Impact of Obesity on Spermatozoa Through Histone Modification

Histones represent a family of highly basic nuclear proteins enriched with lysine and arginine amino acids that play fundamental roles in eukaryotic chromatin organization. These evolutionarily conserved proteins are categorized into five primary classes (H1, H2A, H2B, H3, and H4) and undergo various post-translational modifications, including acetylation, methylation, phosphorylation, and ubiquitination, which dynamically regulate their interactions with DNA. These changes modulate how tightly DNA is packed, thereby regulating the binding of transcription factors and ultimately controlling gene activity [100].

Deshpande et al. demonstrated that diet-induced obesity results in the dysregulation of histone modifications within testicular cells, affecting both transcriptionally activating marks (H3K4me3, H3ac, H4ac) and repressive marks (H3K9me3, H3K27me3). In contrast, obesity of genetic origin selectively alters the levels of acetylated histones [101]. Their findings indicate that the distinct alterations in histone modification levels and the activities of their modifying enzymes in the testis and spermatozoa between diet-induced and genetically induced obesity models may have implications for fertility outcomes. Furthermore, these histone modifications in testicular and sperm cells appear to correlate with variations in white adipose tissue accumulation. In a related investigation, Deshpande et al. observed marked alterations in the expression patterns of key histone-modifying enzymes—including H3K4me3/H3K9me3/H3K27me3-related methyltransferases (HMTs) and demethylases (HDMs), along with H3ac/H4ac-associated acetyltransferases (HATs) and deacetylases (HDACs)—in placental tissue following paternal obesity [102]. Complementary research by Pepin et al. proposes that spermatozoal H3K4me3 may serve as a nutritional status biomarker, potentially transmitting paternal dietary influences to offspring phenotypes through placental mechanisms [103]. Supporting this conceptual framework, Terashima et al. [104] demonstrated that paternal consumption of HFD induces epigenetic reprogramming of developmental histones in sperm, providing mechanistic evidence for histone-mediated transgenerational inheritance.

#### 4.4.3. The Impact of Obesity on Sperm RNA

Environmental factors such as inadequate nutrition, psychological stress, and exposure to toxic substances have been shown to alter the RNA profile of paternal sperm, thereby affecting the phenotype of the offspring [105]. The pathophysiological mechanisms involve multiple epigenetic regulatory processes, including alterations in DNA methylation patterns, histone modifications, and non-coding RNA-mediated gene regulation [106]. Of particular significance, spermatozoa contain a complex repertoire of coding transcripts (mRNAs) and regulatory ncRNAs that play essential roles in facilitating fertilization and orchestrating early embryogenesis [107].

Prolonged obesity in male murine models induces substantial modifications in sperm miRNA expression patterns [108]. Fontelles et al. demonstrated that paternal exposure to HFD resulted in reduced sperm concentrations in offspring compared to control groups maintained on standard diets, accompanied by notable changes in sperm miRNA expression patterns [109]. Interestingly, exercise intervention effectively reversed specific diet-induced miRNA alterations, including three consistently upregulated miRNAs (miR-6538, miR-129-1, and miR-7b) and five downregulated species (miR-143, miR-872, miR-21a, miR-196a-1, and miR-200a). These observations imply that physical activity may contribute to reproductive system reprogramming through epigenetic modulation of key regulatory miRNAs [110]. Furthermore, miR-135b is closely associated with inflammation and embryonic development [111,112]. Collectively, these data indicate that spermatozoa from obese subjects exhibit increased levels of miRNAs related to inflammation and iron metabolism.

A comparative analysis revealed distinct expression profiles of sperm-derived small non-coding RNAs—including microRNAs (miRNAs), PIWI-interacting RNAs (piRNAs), and small nuclear RNA (snRNA) fragments—between obese and lean male subjects. Wang et al. specifically reported dysregulation in several piRNA species, with significant expression differences identified in piR-31115, piR-33044, piR-36378, piR-36707, and piR-57942 [113]. Furthermore, recent studies have identified multiple differentially expressed circular RNAs (circRNAs) in spermatozoa from mice subjected to HFD. Functional analyses have linked these circRNAs to pathways involved in oxidative stress response and the regulation of sperm viability. Notably, testicular spermatozoa from HFD-fed mice exhibited increased circRNA production, whereas epididymal spermatozoa from the same cohort showed a reduced capacity for circRNA generation [51].

### 4.5. Intestinal Microbiota

The gut microbiota plays a crucial role in the development of obesity and metabolic disorders. Moreover, the gut microbiota and its metabolites possess the ability to influence and regulate multiple organs beyond the gastrointestinal tract, particularly the male reproductive system [114]. Hao et al. demonstrated that consumption of an HFD impairs male reproductive function by inducing dysbiosis of the gut microbiota [115]. Interestingly, supplementation with oligosaccharide- and alginate-modified intestinal flora was found to enhance sperm quality and improve fertility outcomes. Conversely, fecal microbiota transplantation from rodents fed an HFD to control animals resulted in spermatogenic abnormalities and a significant reduction in sperm viability [116]. Similarly, MARTinot et al. reported that diets rich in fat or sugar disrupt the equilibrium of gut microbiota, leading to increased endotoxin production [117]. These endotoxins activate inflammatory signaling pathways via Toll-like receptors in testicular blood vessels, weakening endothelial integrity. Consequently, inflammatory mediators infiltrate testicular tissue, facilitating immune cell recruitment and causing spermatogenic damage, which impairs reproductive function [118]. Furthermore, recent investigations have revealed that diet-induced obesity alters the paternal gut microbiome, with these changes being transmitted to offspring [119]. Therefore, modulation of the gut microbiota represents a promising avenue for elucidating the mechanisms underlying the impact of obesity on male fertility.

### 4.6. Oxidative Stress

Wang et al. conducted a comprehensive meta-analysis examining the relationship between obesity and male semen characteristics [120]. Their results revealed that, although sperm concentration and morphological normality were not significantly affected, obese individuals exhibited reductions in seminal volume, total sperm count, progressive motility, and sperm survival rates. These alterations are likely mediated by oxidative stress pathways, which adversely impact both the structural integrity and functional capacity of spermatozoa [121]. Obese men present elevated levels of ROS [122], which can detrimentally influence sperm mitochondrial function and nuclear integrity. Obesity disrupts the physiological balance between ROS and reactive nitrogen species (RNS), leading to oxidative and nitrosative stress that may impair male fertility. Antioxidant treatments aimed at mitigating nitrosative stress may alter sperm chromatin condensation, thereby increasing the susceptibility of DNA to ROS and RNS, which could negatively impact fertilization and early embryonic development [123]. Furthermore, ROS interfere with epigenetic regulatory mechanisms by altering enzyme activities and substrate availability, potentially causing irreversible damage to sperm chromatin structure [124]. Excessive accumulation of ROS beyond physiological thresholds results in oxidative damage to polyunsaturated fatty acids (PUFAs) within the sperm plasma membrane, compromising essential components such as mitochondria and DNA [68]. Consequently, the endogenous antioxidant defense systems of spermatozoa become impaired, leading to reduced vitality, motility, and overall fertility.

Oxidative stress has been recognized as a key factor inducing alterations in mitochondrial function, a condition commonly termed mitochondrial dysfunction [125]. Jing et al. reported that excessive dietary fat intake produces similarly detrimental effects on both the structural and functional integrity of sperm mitochondria, as well as on oxidative stress levels, in humans and mice, ultimately resulting in decreased sperm motility [52]. For instance, elevated cholesterol levels within the spermatozoa of obese men have been implicated in impaired sperm function and premature acrosome reaction [79]. Moreover, studies have demonstrated a positive correlation between the mitochondrial DNA (mtDNA) to nuclear DNA (nDNA) ratio and BMI in obese men [126], with this mtDNA/nDNA ratio also showing a positive association with relative telomere length in this population [127]. The interplay among oxidative stress, dysregulation of autophagy-related gene expression, sperm telomere attrition, and subsequent DNA damage may contribute to the observed decline in sperm quality associated with obesity [128]. Investigating the relationship between sperm mitochondrial activity and telomeric integrity may provide valuable insights into novel mechanisms underlying male reproductive dysfunction.

## 5. Intervention Measures

### 5.1. Weight Loss

Weight reduction has been demonstrated to ameliorate endocrine abnormalities in obese men, evidenced by increased serum testosterone levels and enhanced sexual function [70]. A meta-analysis conducted by Santi D et al., encompassing 12 studies, corroborated that weight loss can improve both qualitative and quantitative sperm parameters, thereby supporting the recommendation of weight reduction for obese male partners exhibiting altered semen profiles in couples attempting conception [129]. Additionally, a notable reduction in sperm DNA fragmentation index was reported following weight loss [129]. Sharma A et al. performed a randomized controlled trial to assess improvements in sperm motility subsequent to dietary interventions involving a low-energy diet and a short-term nutritional regimen [130]. Their findings indicate that modest weight loss achieved through dietary modification may be sufficient to enhance sperm motility in obese men, potentially improving fertility outcomes in couples facing male factor infertility. Presently, clinical guidelines for infertility management have yet to comprehensively incorporate strategies targeting the enhancement of semen quality via weight loss in male partners [131]. Further research is warranted to evaluate the feasibility, clinical efficacy, and cost-effectiveness of publicly accessible dietary intervention programs designed to improve fertility outcomes among infertile couples.

### 5.2. Weight Loss Drugs

Obesity adversely affects male fertility by disrupting the hypothalamic–pituitary–gonadal axis, resulting in altered levels of testosterone and other reproductive hormones [132]. Aromatase inhibitors, including anastrozole, have been utilized as therapeutic interventions for infertility in obese individuals [84]. A meta-analysis and systematic review have demonstrated that treatment with letrozole or anastrozole significantly improves sperm concentration, total sperm count, serum luteinizing hormone, follicle-stimulating hormone, testosterone levels, and the testosterone-to-estradiol ratio, while simultaneously decreasing estradiol levels compared to baseline values [133]. Additionally, letrozole has been reported to enhance sperm concentration and increase the testosterone-to-estradiol ratio in men with oligozoospermia who initially exhibit a normal testosterone-to-estradiol ratio [134]. However, given the potential adverse effects on bone health associated with prolonged use of aromatase inhibitors [135], further large-scale randomized controlled trials are necessary to comprehensively evaluate their long-term safety and therapeutic efficacy.

Obese males often present with reduced testosterone levels; however, this finding alone does not warrant the initiation of testosterone replacement therapy. Administration of exogenous testosterone has been shown to suppress spermatogenesis. Research on hormonal contraceptive methods indicates that most men regain normal sperm production within one year after cessation of treatment. Clomiphene citrate has been recognized as a safe and effective therapeutic option for men aiming to preserve fertility potential [136]. Furthermore, glucagon-like peptide-1 analogs represent a promising alternative treatment, particularly in cases of testosterone deficiency associated with severe obesity [137]. It is advisable to promote weight loss through lifestyle interventions, including dietary modifications and increased physical activity, in all patients presenting with overweight or obesity. For individuals unlikely to achieve resolution of their condition within a reasonable timeframe, a combined therapeutic approach involving testosterone replacement therapy alongside lifestyle modifications may be considered appropriate.

### 5.3. Regulate Intestinal Microbiota

An increased prevalence of Prevotella species in semen has been correlated with oligozoospermia and asthenozoospermia in the context of obesity [114]. Hao et al. were the first to demonstrate that fecal microbiota transplantation utilizing alginate oligosaccharides can beneficially modulate gut microbiota composition, mitigate the decline in semen quality induced by a high-fat diet, and enhance male fertility through favorable alterations in blood and testicular metabolomes [115]. The impairment of vitamin A absorption was primarily attributed to a significant reduction in bile acid concentrations, resulting from disruptions in gut microbiota homeostasis. The study conducted by Zhang et al. elucidated the critical role of vitamin A metabolism within the gut–testis axis. Collectively, these findings suggest that restoring gut microbial balance and normalizing vitamin A metabolic pathways may represent promising therapeutic strategies for male infertility associated with metabolic syndrome [138]. Moreover, the combined administration of probiotics and metformin has been demonstrated to reduce inflammation and oxidative stress while enhancing androgen production, thereby improving testicular spermatogenic function through the restoration of intestinal microbiota equilibrium in high-fat diet-induced murine models [139].

### 5.4. Improve Oxidative Stress

Extensive research has demonstrated that various dietary natural polyphenols, particularly flavonoids derived from fruits, vegetables, and edible plants, significantly influence mitochondrial metabolism and biogenesis, as well as the regulation of ROS homeostasis [140]. For instance, treatment with whole tomato lipid extract has been shown to markedly reduce gonadal adipose tissue mass and oxidative stress, preserve testicular size, improve sperm quality parameters, and maintain the histological integrity of seminiferous tubules. The capacity of these plant-derived bioactive compounds to modulate mitochondrial function likely represents a key mechanism underlying the observed enhancements in male reproductive outcomes [141]. Additionally, administration of marjoram or sage essential oils in obese murine models normalized body weight, sperm count, germ cell apoptosis, and cellular proliferation [142]. Collectively, these findings indicate that dietary natural polyphenols and essential oils may offer a promising therapeutic strategy for addressing obesity-associated male infertility. Their potential to mitigate weight gain, testicular oxidative damage, and apoptotic activity provides valuable insights for the development of novel pharmacological interventions aimed at improving male reproductive function.

## 6. Conclusions

Obesity has experienced a significant increase in prevalence over recent decades, and the detrimental effects it has on the human body are increasingly being highlighted. Body fat distribution patterns represent an important determinant of metabolic and reproductive health risks. In males, excessive adiposity adversely affects multiple aspects of reproductive function, including erectile physiology, spermatogenesis, and transgenerational health outcomes. These effects are mediated through interconnected pathways involving hypothalamic–pituitary–gonadal axis dysregulation, systemic and local inflammatory responses, scrotal hyperthermia, epigenetic reprogramming, gut microbiome alterations, and oxidative stress accumulation. Notably, microRNAs in germ cells are restored through physical exercise. This review proposed improved strategies and treatment approaches for preventing and managing obesity-related low male fertility. Notably, implementing dietary improvements, engaging in regular exercise, and adopting healthier eating habits represent effective approaches for reducing obesity and its detrimental impact on male reproductive function.

This review recognizes several limitations. In particular, it has predominantly presented recent relevant studies without performing a systematic analysis of their conflicting outcomes. Further research is necessary to substantiate these findings; nevertheless, this review provides a basis for developing various research avenues. Additionally, it delineates optimized intervention strategies and therapeutic approaches for managing obesity-related male subfertility.

## Figures and Tables

**Figure 1 biomedicines-13-02054-f001:**
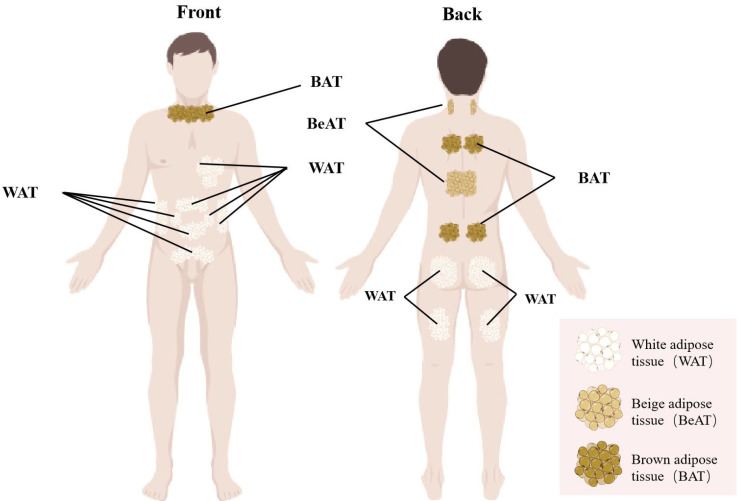
The distribution of adipose tissue in human. WAT is predominantly distributed in the abdominal and thoracic regions, as well as within the gluteofemoral muscles. In contrast, BAT is primarily localized in the clavicular, scapular, and paraspinal areas. Beige adipose tissue is chiefly located on the posterior aspect of the neck and subcutaneously along the lateral sides of the spine.

**Figure 2 biomedicines-13-02054-f002:**
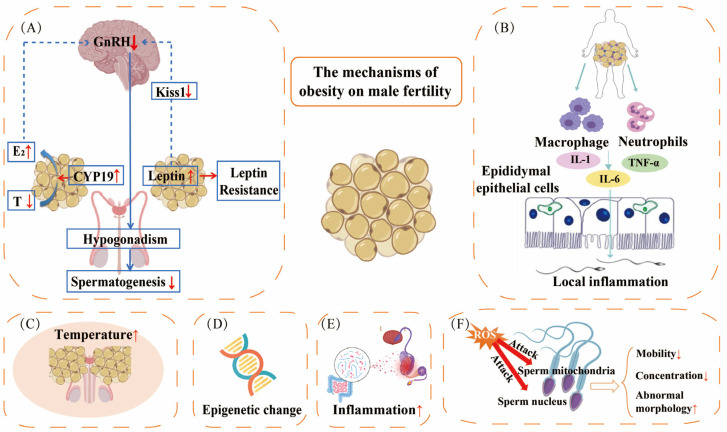
The impact of obesity on male fertility. (**A**) Obesity has significant implications for male fertility, primarily through several physiological mechanisms. Adipose tissue elevates CYP19 in adipocytes, enhancing testosterone (T) conversion to estradiol (E2). This negative feedback on the hypothalamic–pituitary–testicular (HPT) axis potentially causes gonadal dysfunction. In addition, long-term elevated levels of leptin in obese males lead to leptin resistance, decreased Kiss expression, and reduced stimulation of GnRH neurons. A decrease in GnRH leads to male hypogonadism and spermatogenesis disorders. (**B**) Macrophages, neutrophils, and inflammatory mediators in adipose tissue infiltrate the epididymal lumen, leading to increased cytokine (IL-1, IL6, and TNF-α) expression and epithelial cell apoptosis, which in turn affect sperm maturation and the fertilization process. (**C**) Obesity is associated with increased local temperature in the scrotum, which adversely affects male fertility. (**D**) The quality of sperm is also compromised by obesity through epigenetic modifications. (**E**) Obesity disrupts the gut microbiota, leading to testicular inflammation and subsequent damage to sperm quality. (**F**) Elevated levels of reactive oxygen species (ROS) can adversely affect sperm mitochondria and nuclei, leading to decreased sperm motility and concentration, as well as an increased prevalence of abnormally shaped sperm.

**Table 1 biomedicines-13-02054-t001:** The distinction between WAT and BAT.

	WAT	BAT
Cellular structure	Unilocular lipid droplets, relatively sparse mitochondria [27].	Multilocular lipid droplets, and more mitochondria [32].
Distribution	Thighs, buttocks, lower abdomen, and pubic area, and visceral fat [28].	Cervical, perirenal, suprarenal, and cardiothoracic areas, particularly surrounding the aorta and mediastinal structures [31].
Main functions	Energy storage [4].	Heat production (non-shivering thermogenesis) [33].
Relationship with obesity	Dysregulation of WAT expansion and function [35], excessive fat accumulation [36].	Increase energy expenditure and protect against obesity [37].

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
