# Peer review of "Progress in Investigating the Impact of Obesity on Male Reproductive Function"

_biomedicines, 2025, doi:10.3390/biomedicines13092054_

Round 1

Reviewer 1 Report

Comments and Suggestions for Authors

General comments

The topic is unique and worthy of researching, as this review aimed to review the classification and distribution of adipose tissue in obesity, the impact of obesity on male fertility, and the potential mechanisms through which obesity affects male reproductive health,  thereby offering insights into the prevention and treatment of obesity-related male fertility issues.. The deduced conclusions based on the research methods/cases are enough and tenable. Regarding, the progress that had been made compared with the current research results; this Review highlights and proposed  that improved strategies and treatment approaches for preventing and managing obesity-related low male fertility. Notably, implementing dietary improvements, engaging in regular exercise, and adopting healthier eating habits represent effective approaches for reducing obesity and its detrimental impact on male reproductive function. This review outlines optimized intervention strategies and therapeutic modalities for addressing obesity-associated male subfertility.

Strengths and weaknesses

The abstract is informative and reflect the body of the paper. The introduction provides sufficient background information for readers in the immediate field to understand the problem/hypotheses. The text is well arranged and the logic, the related concepts introduced clearly and the readability is sufficient. The discussion and theoretical analysis in this article is good. The reference section is informative and accurate.

Suggestions for improvement

Paper is good, but need some language. The reference section needs to be updated.

Comments on the Quality of English Language

Paper is good, but need some language. 

Author Response

Thank you for pointing this out. We agree with this comment. We have revised and adjusted the language throughout the text and marked the changes in blue. References 19-20 in the introduction have been updated, and the additional content is all new literature.

Reviewer 2 Report

Comments and Suggestions for Authors

The review by Kang et al. addresses an interesting and timely topic with clear relevance. However, there are several issues that need to be addressed to improve the overall quality, structure, and clarity of the manuscript. Below are my detailed comments:

General Comments

  • The manuscript would benefit from a more critical perspective. Currently, it presents findings and literature in a descriptive manner without sufficient evaluation of study limitations or conflicting evidence.

  • The format of the author list is not correct. Some names are not capitalized, and a tab should be inserted between the author names and their affiliations. Additionally, several authors lack affiliations.

  • Keywords: The term “male infertility” should be added to the list of keywords for better indexing and relevance.

Formatting Issues:

Add tabs where missing:

  • Line 46, before reference [4]

  • Figure 2 (Line 226), before the word "obesity"

  • Line 306, after reference [67]

  • Line 356, before reference [79]

  • Line 433, before reference [98]

Line-specific Corrections:

  • Line 132: Remove the double dot after reference [31].

  • Line 158: The abbreviation HDF appears for the first time—please spell out the full term.

  • Line 166: The term Ldlr.Leiden mice is unclear—please verify and revise if necessary.

  • Line 193: Remove the dot before reference [52].

  • Line 228: Gene names should be italicized.

  • Line 263: Remove the dot before reference [63].

  • Line 420: Remove the dot after reference [93].

  • Line 430: Remove the dot before reference [96].

  • Line 440: Remove the dot before reference [99].

  • Line 452: Remove the dot after reference [102].

  • Line 457: Remove the dot after reference [104].

Introduction Section:

  • Consider adding background information on male infertility, including its definition, epidemiological data, and major causes, to set the context for the review.

Figure 1:

  • The beige box on the right contains several errors. For example:

    • "Dissue" should be corrected to "tissue"

    • "Brite" should likely be "beige"

Section 4.3:

  • This section lacks references. For example, in line 319, the study by Garolla et al. is mentioned, but no citation is provided. Please ensure all claims are properly supported with references.

Section 4.5:

  • This section would benefit from a brief introductory paragraph discussing the relationship between the gut microbiome, obesity, and male infertility before delving into specific findings.

Section “Other”:

  • This section is somewhat unclear in its focus. The majority of the content revolves around oxidative stress, which is a well-established mechanism linking obesity and male infertility. Please see these studies: 

  • Martínez-Martínez, E., & Cachofeiro, V. (2022). Oxidative Stress in Obesity. Antioxidants (Basel, Switzerland), 11(4), 639. https://doi.org/10.3390/antiox11040639,
  • Kaltsas, A., Markou, E., Kyrgiafini, M. A., Zikopoulos, A., Symeonidis, E. N., Dimitriadis, F., Zachariou, A., Sofikitis, N., & Chrisofos, M. (2025). Oxidative-Stress-Mediated Epigenetic Dysregulation in Spermatogenesis: Implications for Male Infertility and Offspring Health. Genes, 16(1), 93. https://doi.org/10.3390/genes16010093
  • I recommend splitting this section into two:

    • One subsection dedicated to oxidative stress, expanded with additional literature 

    • A second subsection focused on mitochondrial regulation and its implications in the context of male infertility

  • This reorganization would enhance the clarity and scientific depth of the discussion.

Concluding Section (Lines 494–498):

    • Statements such as “This review proposed improved strategies and treatment approaches...” and “This review outlines optimized intervention strategies...” are not fully supported, as the manuscript currently does not provide a critical analysis or practical guidelines.

    • I recommend adding a new subsection before the conclusions that:

      • Discusses current limitations in the literature

      • Suggests future research directions

      • Proposes how existing findings could be translated into clinical applications

    • Alternatively, if the above cannot be addressed, these overstated sentences should be removed. However, the above sections could improve the overall manuscript.

Data Availability Statement:

    • Please clarify this statement. As this is a review article, it is unclear what "data" are available upon request. If no original data are presented, this statement may not be necessary.

Comments on the Quality of English Language

The manuscript contains numerous typographical errors, inconsistencies in formatting, and minor issues with syntax and grammar. A thorough English language and style editing is required.

Author Response

Response 2: Thank you very much for your comments, which are very helpful to our article. Below, I will respond to your comments one by one:

1.This article still has limitations. Additional information regarding the limitations of this article can be found in lines 596-599 of the revised manuscript, which have been highlighted in yellow.

2.The author names and affiliations have been modified and highlighted in yellow.

The keyword ‘male infertility’ has been added and highlighted in yellow. It can be seen in line 34.

3.Formatting Issues have been revised throughout the text in accordance with the comments. Since the overall format has been corrected, no additional annotations have been made.

4.The introduction section has been updated with background information on male infertility, including its definition, epidemiological data, and primary causes. This information has been highlighted in yellow and can be found in lines 36-45.

5.The spelling error in Figure 1 has been corrected.

6.The references in Section 4.3 have been supplemented and highlighted in yellow. They can be found in lines 309-325. Other sections have also been checked.

7.The introduction has been added to Section 4.5 and can be found in lines 443-446 highlighted in yellow.

8.Based on your comments regarding Section 4.6 and the references provided, I have revised the title of this section to ‘Oxidative Stress.’ Since mitochondrial damage observed in obesity is associated with oxidative stress, this section first discusses the relationship between oxidative stress and obesity, followed by an examination of the connection between oxidative stress-induced mitochondrial dysfunction and obesity. Relevant literature has also been added. The modifications to this section have been highlighted in yellow and can be viewed on lines 474–488 and 494–496.

9.Based on your suggestion to add a sub-section, I have added a fifth section on intervention measures and presented the improvement in male fertility after intervention measures, providing treatment ideas for clinical obesity-induced male infertility. The added section has been highlighted in yellow and can be found on lines 500-580.

10.As this article is a review and does not present any original data, the data availability statement has been deleted and highlighted in yellow, which can be seen in line 621.

11.Regarding language issues, we have revised the entire text, and the revised language issues are highlighted in blue.

Reviewer 3 Report

Comments and Suggestions for Authors

Dear author, 

this review is related to important aspect of human health and scientific community could benefit from it. 

Those are my comments in order to improve this review.

  1. The adipose tissue classification and distribution should be further explained specifically related to male reproduction.
  • Line 74-75: This state indicate that WAT and BAT have same metabolic function (role). I suggest presenting all difference between WAT and BAT in form of Table. 
  • Line 86: “…brown adipocytes contain larger mitochondria than their white adipocyte...”. Please correct this. BAT have more (number) mitochondria than WAT, not larger (size) mitochondria.
  • Line 106-110: Please add reference.
  • Line 113: Figure 1 should represent specific distribution of adipose tissue in men. Mark EAT also, because you detail explain it later in text (you can add additional figure 1b with larger presentation of fat related to male reproductive organs). Also, it is not clear what red rectangle presents.
  • Line 133-134: No need to note characteristics of rodents if there is no additional explanation and differences between humans.

2. In the section 3.1. The Impact of Obesity on Spermatogenesis and Sperm Quality only viability is mentioned related to sperm quality. This had to be improved by mentioning other factors related to sperm quality like morphology, concentration, volume, pH…).

3. Instead of using term “Males with Obesity” which is not common in scientific writing use Male Obesity or Obesity in Males.

4. Figure 2 description is too large. Detailed explanation should be in text with references.

5. In the section 4.3. Elevated Localized Heat in the Reproductive System references should be added

6. It is indicated that one of the mail goals of this review is “offering insights into the prevention and treatment of obesity-related male fertil-31 ity issues”. This part is complitly missing.

Generally, in text please check grammar and references citation (some are citated like Line 133 – Chu et al.[34] reported, also Line 160, 163, 165, 174, 187, 210…etc. should be citated in why that number of references goes at the end of sentence).

Author Response

Thank you very much for your comments, which are very helpful to our article. Below, I will respond to your comments one by one:

I have created Table 1 to show the differences between WAT and BAT, and added relevant literature.

BAT has more mitochondria than WAT, not larger mitochondria. I have corrected the sentence and highlighted it in yellow. You can see it in line 95.

References for lines 106-110 have been added and highlighted in yellow. After modification, they can be seen in line 120.

Figure 1 in line 113 has been modified and indicator lines have been added. The fat distribution in men is mainly as shown in the figure, with fat blocks added around the abdomen and waist. The fat in this area and the heart area is visceral fat. Therefore, Figure 1b was not drawn separately. It can be seen in line 111.

Figure 1 in line 113 has been modified and indicator lines have been added. The fat distribution in men is mainly as shown in the figure, with fat blocks added around the abdomen and waist. The fat in this area and the heart area is visceral fat. Therefore, Figure 1b was not drawn separately. It can be seen in line 111.

I have chosen to retain the description of rodent fat characteristics in lines 133–134. This is because the eWAT described below is a unique fat depot in the male reproductive system of rodents, enveloping the epididymis and extending into the peritoneal cavity. Unlike subcutaneous or visceral fat, it is located near the testicles and influences reproductive and metabolic processes. Therefore, Figure 1 does not label eWAT. The description of this fat reservoir is intended to illustrate the association between fat and male fertility.

3.1 Additional information on other parameters related to obesity causing a decline in male sperm quality has been added and can be found in lines 179-182.

The wording ‘Males with Obesity’ has been changed to ‘Male Fertility,’ as can be seen in line 161.It has been highlighted in yellow.

The description in Figure 2 has been simplified. The previous description has already been reflected in each section, so there is no need for additional supplementation.

The references in Section 4.3 have been supplemented and highlighted in yellow. They can be found in lines 309-325.

I have added a fifth section on intervention measures and presented the improvement in male fertility after intervention measures, providing treatment ideas for clinical obesity-induced male infertility. The added section has been highlighted in yellow and can be found on lines 500-579.

Round 2

Reviewer 2 Report

Comments and Suggestions for Authors

Thank you for your thoughtful and comprehensive revisions to the manuscript. I appreciate the effort you have made to address the previous comments. The revised version is substantially improved in terms of clarity, structure, and scientific rigor.

Author Response

Thank you for the opportunity to revise our manuscript  for consideration. We sincerely appreciate the time and effort invested by you in evaluating our work. Their insightful comments and constructive suggestions have been invaluable in strengthening our manuscript.

Reviewer 3 Report

Comments and Suggestions for Authors

Dear authors, 

paper is improved, so I can recommend acceptance in this form.

Author Response

We are pleased that you are satisfied with the changes made based on the previous feedback. Thank you again for your valuable comments on our manuscript.